# Stieltjes Transforms and *R*-Transforms Associated with Two-Parameter Lambert–Tsallis Functions

**DOI:** 10.3390/e25060858

**Published:** 2023-05-27

**Authors:** Hideto Nakashima, Piotr Graczyk

**Affiliations:** 1The Institute of Statistical Mathematics, Midori-cho 10-3, Tachikawa, Tokyo 190-8562, Japan; 2Laboratoire de Mathématiques LAREMA, Université d’Angers, 2 boulevard Lavoisier, CEDEX 01, 49045 Angers, France; piotr.graczyk@univ-angers.fr

**Keywords:** random matrices, eigenvalue distributions, Stieltjes transformations, R-transformations, Lambert–Tsallis functions

## Abstract

In this paper, we study a two-parameter family of Stieltjes transformations related to holomorphic Lambert–Tsallis functions, which are a two-parameter generalization of the Lambert function. Such Stieltjes transformations appear in the study of eigenvalue distributions of random matrices associated with some growing statistically sparse models. A necessary and sufficient condition on the parameters is given for the corresponding functions being Stieltjes transformations of probabilistic measures. We also give an explicit formula of the corresponding *R*-transformations.

## 1. Introduction

This paper is a continuation of a previous paper [1] dealing with high-dimensional asymptotics of eigenvalue distributions of Gaussian and covariance matrices related to growing statistical sparse models. The paper [1] gives an explicit formula of Stieltjes transformations of the limits of empirical eigenvalue distributions on some Wishart-type ensembles. Asymptotics of empirical eigenvalue distributions are a classical topic of the random matrix theory (RMT). There are numerous interactions of RMT with important areas of modern multivariate statistics: high-dimensional statistical inference, estimation of large covariance matrices, principal component analysis (PCA), time series and many others; see the review papers by Diaconis [2] [Section 2], Johnstone [3], Paul and Aue [4], Bun et al. [5], and the book by Yao et al. [6] and references therein.

Stieltjes transformations and *R*-transformations are fundamental objects in Random Matrix Theory and Free Probability Theory, and basic tools for investigating probabilistic measures (cf. Voiculescu [7], Mingo and Speicher [8], and references therein). In particular, *R*-transformations admit some additive property with respect to a sum of two random matrices under some conditions. We study *R*-transformations associated with Wishart-type ensembles considered in [1] in detail in Section 3.

The previous paper [1] tells us that Stieltjes transformations of limiting density functions with respect to Wishart-type ensembles in [1] are described using a two-parameter generalization $W_{\kappa ,\gamma }$ of the Lambert *W* function, called the Lambert–Tsallis function. It is an inverse function of the product of a linear fraction of *z* and the Tsallis *q*-exponential function; see (Equation 1) for a definition. We note that the Tsallis *q*-exponential function is now actively studied in information geometry [9,10] and physics [11,12,13]. There were several attempts to generalize the Lambert function in many directions. In Mezö and Baricz [14], *z* before $e^{z}$ is replaced by rational function of *z* (see also [15,16]) and in da Silva and Ramos [11], $e^{z}$ is replaced by the Tsallis *q*-exponential function. The matrix Lambert *W* function is considered in [17,18].

In Section 4, we investigate Lambert–Tsallis functions in detail to obtain a complete set of parameters such that the corresponding functions are Stieltjes transformations of probabilistic measures. The proofs are in Section 5.

The results of this paper should be useful in further applications of Lambert–Tsallis functions in the Random Matrix Theory, in particular Theorem 3 and Corollary 2, and also the open problem formulated below it.

This paper can be read independently of the paper [1], if the reader admits Theorem 1 below.

## 2. Preliminaries

We begin this paper by fixing basic notions. The sets of real numbers and complex numbers are described R and C, respectively. We set C+:=z∈C;Imz>0. We denote by Mat(n,m;R) the space of n×m matrices with real coefficients. When m=n, we simply write Mat(n,R). For a matrix X∈Mat(n,m;R), its transpose matrix is denoted by $X^{⊤}$.

For a non-zero real number κ, we set
$$\exp_{\kappa }\left(z\right):={\left(1+\frac{z}{\kappa }\right)}^{\kappa }\;\left(1+\frac{z}{\kappa }\in C\backslash R_{\leq 0}\right),$$
where we take the main branch of the power function when κ is not an integer. If $\kappa =\frac{1}{1-q}$, then it is exactly the so-called Tsallis *q*-exponential function(cf. [9,10]). For the sake of simplicity, we use the parameter κ rather than *q*. By virtue of $\lim_{\kappa \to \infty }\exp_{\kappa }\left(z\right)=e^{z}$, we define $\exp_{\infty }\left(z\right)=e^{z}$.

For two real numbers κ,γ such that κ≠0, we introduce a holomorphic function $f_{\kappa ,\gamma }\left(z\right)$, which we call *generalized Tsallis function*, by
(1)$$f_{\kappa ,\gamma }\left(z\right):=\frac{z}{1+\gamma z}\exp_{\kappa }\left(z\right)\;\left(1+\frac{z}{\kappa }\in C\backslash R_{\leq 0}\right).$$
Analogously to Tsallis *q*-exponential, we also consider $f_{\infty ,\gamma }\left(z\right)=\frac{ze^{z}}{1+\gamma z}$(z∈C). In particular, $f_{\infty ,0}\left(z\right)=ze^{z}$. Since it is easily verified that $f_{\kappa ,\gamma }^{\prime }\left(0\right)\neq 0$, the function $f_{\kappa ,\gamma }\left(z\right)$ has an inverse function $w_{\kappa ,\gamma }$ in a neighborhood of z=0.

Such a function appears in the study of Wishart-type random matrices in the previous paper [1]. Let us consider the limit of eigenvalue distributions of matrices of the form
(2)
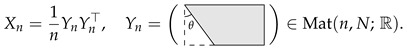

Here, we constraint that the ratio $\frac{N}{n}$ converges to a constant *c* as *n* grows, and the angle parameter θ is independent of *n*. Set γ=1−c and $\kappa =\frac{1}{1-\tan\theta }$. Then, the Stieltjes transformations corresponding to the limiting eigenvalue distributions are given as follows.

**Theorem** **1**([1] [Proposition 4.7])**.**
*The Stieltjes transformation S(z) corresponding to the limiting eigenvalue distribution with respect to $X_{n}$ is given as*
$$S\left(z\right)=S_{\kappa ,\gamma }\left(z\right)=-1-\frac{1}{zW_{\kappa ,\gamma }\left(-\frac{1}{z}\right)}-\frac{\gamma }{z}=-1+\exp_{\kappa }\left(W_{\kappa ,\gamma }\left(-\frac{1}{z}\right)\right)\;\left(z\in C^{+}\right).$$
*Here, $W_{\kappa ,\gamma }$ is the Lambert–Tsallis function, which is the holomorphic extension of the inverse function $w_{\kappa ,\gamma }$ of the generalized Lambert function $f_{\kappa ,\gamma }$. In the current situation where the parameters satisfy $\frac{1}{\kappa }\geq \gamma$, such an extension is unique.*

This theorem tells us that we have a two-parameter family ${\left\{S_{\kappa ,\gamma }\left(z\right)\right\}}_{1/\kappa \geq \gamma }$ of Stieltjes transformations related to Wishart-type random matrices. Then, it is natural to raise the question of whether the domain of parameters can be extended or not. This is half of the aim of this paper.

The other half is to investigate the corresponding *R*-transformations in detail. By Theorem 1, we can obtain the formula of the corresponding *R*-transformations as in Proposition 4.7 of [1]. *R*-transformations relate to the theory of free probability, and they have interesting properties such as homomorphism with respect to asymptotically free elements. We first work on this topic.

## 3. *R*-Transforms

We review here the basic properties of *R*-transformations. For a probability measure μ, we let S(z) be its Stieltjes transformation. Its *R*-transformation R(z) is defined as
$$S\left(R\left(z\right)+\frac{1}{z}\right)=-z$$
as a formal power series at first, and it is shown that it can be defined as an analytic function on a domain in $C^{+}$. In particular, if μ has compact support, then it is known that the corresponding *R*-transformation R(z) is analytic around z=0.

In this section, we consider the limiting eigenvalue distributions μ with respect to Wishart matrices of the form (Equation 2). We have calculated the support of μ explicitly in Theorem 18 of the previous paper [1]. It tells us that μ has compact support so that the corresponding *R*-transformations are analytic in a neighborhood of z=0. To state formulas, we recall that the inverse function of the Tsallis exponential $\exp_{\kappa }\left(z\right)$ is given as
$$\log^{\langle 1/\kappa \rangle }\left(z\right)=\frac{z^{1/\kappa }-1}{1/\kappa }=\kappa \left(z^{1/\kappa }-1\right)\;\left(z\in C\backslash R_{\leq 0}\right).$$

**Theorem** **2.**
*The R transformation $R_{\kappa ,\gamma }\left(z\right)$ corresponding to $S_{\kappa ,\gamma }\left(z\right)$ is given as*

$$R_{\kappa ,\gamma }\left(z\right)=-\frac{1}{z}-\frac{\gamma }{1-z}-\frac{1}{\left(1-z\right)\log^{\langle 1/\kappa \rangle }\left(1-z\right)}\;\left(1-z\in C\backslash R_{\leq 0}\right).$$

*If we set $g\left(z\right)={\left(1-z\right)}^{\gamma }\cdot \frac{\log^{\langle -1/\kappa \rangle }\left(1-z\right)}{z}$, then the above equation can be reformulated as*

$$R_{\kappa ,\gamma }\left(z\right)=-\frac{1}{z}-\frac{\gamma }{1-z}+\frac{{\left(\log^{\langle -1/\kappa \rangle }\left(1-z\right)\right)}^{\prime }}{\log^{\langle -1/\kappa \rangle }\left(1-z\right)}=\frac{g^{\prime }\left(z\right)}{g(z)}.$$



**Remark** **1.**
*The first equation is given in Proposition 4.7 in [1], whereas the second seems to be new. We also give a proof of the first equation for the reader’s convenience.*


**Proof.** The *R*-transformations and the Stieltjes transformations relate in the formula $R\left(z\right)=S^{-1}\left(-z\right)-\frac{1}{z}$. Here, $S^{-1}\left(z\right)$ stands for the inverse function of S(z). For a Stieltjes transformation S(z), it is known that its inverse exists for *z* such that Imz is large enough, and it can be analytically continued to $C^{+}$. Thus, we first calculate the inverse of $S_{\kappa ,\gamma }\left(z\right)$. Assume that |z| is large enough. Then, since $S\left(z\right)=O\left(\frac{1}{z}\right)$, we see that |S(z)| is enough small. By the second equality of Theorem 1, we have
$$S_{\kappa ,\gamma }\left(z\right)+1=\exp_{\kappa }\left(W_{\kappa ,\gamma }\left(-\frac{1}{z}\right)\right)\Leftrightarrow W_{\kappa ,\gamma }\left(-\frac{1}{z}\right)=\log^{\langle 1/\kappa \rangle }\left(1+S_{\kappa ,\gamma }\left(z\right)\right).$$
By the definition of $f_{\kappa ,\gamma }$, we obtain
$$-\frac{1}{z}=f_{\kappa ,\gamma }\left(\log^{\langle 1/\kappa \rangle }\left(1+S_{\kappa ,\gamma }\left(z\right)\right)\right)\Leftrightarrow z=\frac{1}{f_{\kappa ,\gamma }\left(\log^{\langle 1/\kappa \rangle }\left(1+S_{\kappa ,\gamma }\left(z\right)\right)\right)},$$
which implies the following formula for |z| enough small:
$$S_{\kappa ,\gamma }^{-1}\left(z\right)=\frac{1}{f_{\kappa ,\gamma }\left(\log^{\langle 1/\kappa \rangle }\left(1+z\right)\right)}.$$
Thus, we arrive at
$$\begin{array}{ccc} R_{\kappa ,\gamma }\left(z\right) & = & S_{\kappa ,\gamma }^{-1}\left(-z\right)-\frac{1}{z}=-{\left(\frac{\log^{\langle 1/\kappa \rangle }\left(1-z\right)}{1+\gamma \log^{\langle 1/\kappa \rangle }\left(1-z\right)}\times \exp_{\kappa }\left(f_{\kappa ,\gamma }\left(\log^{\langle 1/\kappa \rangle }\left(1-z\right)\right)\right)\right)}^{-1}-\frac{1}{z}\\ & = & -\frac{1+\gamma \log^{\langle 1/\kappa \rangle }\left(1-z\right)}{\left(1-z\right)\log^{\langle 1/\kappa \rangle }\left(1-z\right)}-\frac{1}{z}=-\frac{1}{z}-\frac{\gamma }{1-z}-\frac{1}{\left(1-z\right)\log^{\langle 1/\kappa \rangle }\left(1-z\right)}. \end{array}$$
Next, we consider the second assertion. Since ${\left(\log^{\langle \alpha \rangle }\left(w\right)\right)}^{\prime }=w^{\alpha -1}$ for α∈R, we have
$$\frac{1}{w\log^{\langle 1/\kappa \rangle }\left(w\right)}=\frac{1}{\kappa w(w^{\frac{1}{\kappa }}-1)}=\frac{w^{-\frac{1}{\kappa }-1}}{\kappa w\left(w^{\frac{1}{\kappa }}-1\right)\cdot w^{-\frac{1}{\kappa }-1}}=\frac{w^{-\frac{1}{\kappa }-1}}{-\kappa (w^{-\frac{1}{\kappa }}-1)}=\frac{{\left(\log^{\langle -1/\kappa \rangle }\left(w\right)\right)}^{\prime }}{\log^{\langle -1/\kappa \rangle }\left(w\right)}.$$
Thus, by setting w=1−z, we obtain
$$R_{\kappa ,\gamma }\left(z\right)=-\frac{1}{z}-\frac{\gamma }{1-z}-\frac{-1\cdot {\left(\log^{\langle -1/\kappa \rangle }\left(1-z\right)\right)}^{\prime }}{\log^{\langle -1/\kappa \rangle }\left(1-z\right)}=-\frac{1}{z}+\frac{{\left({\left(1-z\right)}^{\gamma }\right)}^{\prime }}{{\left(1-z\right)}^{\gamma }}+\frac{{\left(\log^{\langle -1/\kappa \rangle }\left(1-z\right)\right)}^{\prime }}{\log^{\langle -1/\kappa \rangle }\left(1-z\right)}.$$
Noticing $-\frac{1}{z}=\frac{-1/z^{2}}{1/z}=\frac{{\left(1/z\right)}^{\prime }}{1/z}$, we obtain
$$R_{\kappa ,\gamma }\left(z\right)=\frac{g^{\prime }\left(z\right)}{g(z)},\;\mathrm{where}\;g\left(z\right)={\left(1-z\right)}^{\gamma }\cdot \frac{\log^{\langle -1/\kappa \rangle }\left(1-z\right)}{z}.$$
We have shown that the assertions are valid for z in a neighborhood of z=0. On the other hand, we know that *R*-transformations of μ are analytic around z=0 so that we have proved this theorem by the identity theorem for analytic functions. □

The following corollary is a direct consequence of Theorem 2.

**Corollary** **1.**
*In the setting of (Equation 2), we decompose $Y_{n}$ as $Y_{n}=T_{n}+M_{n}$, where $T_{n}$ corresponds to the triangular part of $Y_{n}$, and $M_{n}$ the rectangular part. Then, $X_{n}$ can be viewed as a sum of $\frac{1}{n}T_{n}T_{n}^{⊤}$ and $\frac{1}{n}M_{n}M_{n}^{⊤}$. Let $R_{\mathrm{trap}}\left(z\right)$ (resp. $R_{\mathrm{trig}}\left(z\right)$ and $R_{\mathrm{rect}}$) be the R-transformation with respect to $X_{n}$ (resp. $\frac{1}{n}T_{n}T_{n}^{⊤}$ and $\frac{1}{n}M_{n}M_{n}^{⊤}$). Then, one has*

$$R_{\mathrm{trap}}\left(z\right)=R_{\mathrm{trig}}\left(z\right)+R_{\mathrm{rect}}\left(z\right).$$



**Proof.** Suppose that the aspect ratios of $T_{n}$ and $M_{n}$ are α and *c*, respectively. Then, the shape parameters γ and κ of $Y_{n}$ are given as
$$\gamma =1-c-\alpha ,\;\kappa =\frac{1}{1-\alpha }.$$
By Theorem 2, the *R*-transformations $R_{\mathrm{trap}}\left(z\right)$ and $R_{\mathrm{trig}}\left(z\right)$ are given, respectively, as
$$\begin{array}{ccc} R_{\mathrm{trap}}\left(z\right) & = & -\frac{1}{z}-\frac{1-c-\alpha }{1-z}-\frac{1}{\left(1-z\right)\log^{\langle 1/\kappa \rangle }\left(1-z\right)}\\ R_{\mathrm{trig}}\left(z\right) & = & -\frac{1}{z}-\frac{\alpha }{1-z}-\frac{1}{\left(1-z\right)\log^{\langle 1/\kappa \rangle }\left(1-z\right)}, \end{array}$$
and $R_{\mathrm{rect}}\left(z\right)$ is just the *R*-transformation with respect to the Marchenko–Pastur law of parameter *c* so that
$$R_{\mathrm{rect}}\left(z\right)=\frac{c}{1-z}.$$
Thus, it is now obvious that we have $R_{\mathrm{trap}}\left(z\right)=R_{\mathrm{trig}}\left(z\right)+R_{\mathrm{rect}}\left(z\right)$. □

Although *R*-transformations have some additive property on a sum of two probabilistic measures, this corollary is not trivial because we do not know whether or not triangle Wishart ensembles and Wishart ensembles are asymptotically free. In fact, we have a counterexample of such a decomposition formula as follows. We first recall that the *R*-transformation of Marchenko–Pastur law with parameter *c* is given as
$$R_{\mathrm{rect}}\left(z\right)=\frac{c}{1-z}.$$Let us set (κ,γ)=(−1,−1). In this case, the corresponding matrices $Y_{n}$ are a kind of triangular matrices in Mat(n,2n;R). Since an ordering of column vectors in $Y_{n}$ does not affect the resultant, the matrices $Y_{n}$ can be viewed as two copies of triangular matrices, that is,
$$Y_{n}=\left(T_{1}\;T_{2}\right)\;\mathrm{so}\;\mathrm{that}\;X_{n}=\frac{1}{n}Y_{n}Y_{n}^{⊤}=\frac{1}{n}\left(T_{1}T_{1}^{⊤}+T_{2}T_{2}^{⊤}\right),$$
where $T_{1}$ and $T_{2}$ are upper triangular matrices in Mat(n,R). By Theorem 2, we see that the *R*-transformation corresponding to $X_{n}$ is given as
$$R_{\kappa ,\gamma }\left(z\right)=R_{-1,-1}\left(z\right)=-\frac{1}{z}+\frac{1}{1-z}-\frac{1}{\left(1-z\right)\left(1-\frac{1}{1-z}\right)}=\frac{1}{1-z}.$$This shows that the limit of density functions of the sum of two triangle Wishart ensembles is equal to that of MP laws with parameter c=1:$$“\mathrm{the}\;\mathrm{limiting}\;\mathrm{density}\;\mathrm{of}\;T_{1}T_{1}^{⊤}+T_{2}T_{2}^{⊤}”=“\mathrm{the}\;\mathrm{limiting}\;\mathrm{density}\;\mathrm{of}\;MM^{⊤}”,$$
where $T_{1},T_{2}$ are triangular matrices and *M* is a square random matrix (see Figure 1). On the other hand, Theorem 2 also tells us that the *R*-transformation with respect to $\frac{1}{n}T_{i}T_{i}^{⊤}$ is given as
$$R_{\mathrm{trig}}^{\left(i\right)}\left(z\right)=-\frac{1}{z}-\frac{1}{\left(1-z)\log(1-z\right)}.$$We have thus confirmed that
$$R_{-1,-1}\left(z\right)\neq R_{\mathrm{trig}}^{\left(1\right)}\left(z\right)+R_{\mathrm{trig}}^{\left(2\right)}\left(z\right),$$
which shows that two triangle Wishart matrices are not asymptotically free. Recall that if two random matrices are asymptotically free, then the corresponding *R*-transformations admit an additivity property, that is, the *R*-transformation corresponding to their sum is equal to the sum of their *R*-transformations (cf. Chapter 4 of [8], see also [7]).

## 4. A Two-Parameter Family of Stieltjes Transforms

In this section, we investigate the parameter domain of $\left\{S_{\kappa ,\gamma }\right\}$ such that $S_{\kappa ,\gamma }\left(z\right)$ is a Stieltjes transformation of a probability measure. To do so, we first give an explicit definition of Lambert–Tsallis functions.

Let $D(f_{\kappa ,\gamma })$ be the domain of $f_{\kappa ,\gamma }$, i.e., if κ is integer then $D\left(f_{\kappa ,\gamma }\right)=C\backslash \left\{-\frac{1}{\gamma }\right\}$, and if κ is not integer, then $D\left(f_{\kappa ,\gamma }\right)=C\backslash \left\{x\in R;\;1+\frac{x}{\kappa }\leq 0,\;\mathrm{or}\;x=-\frac{1}{\gamma }\right\}.$

Let z=x+yi∈C and we set $\theta \left(x,y\right):=\mathrm{Arg}\left(1+\frac{z}{\kappa }\right)$ for finite κ≠0, where Arg(w) stands for the principal argument of *w*; −π<Arg(w)≤π. Since we now take the main branch of the power function, we have for finite κ≠0
$$f_{\kappa ,\gamma }\left(z\right)=\frac{|1+\frac{z}{\kappa }{|}^{\kappa }}{{\left(1+\gamma x\right)}^{2}+\gamma^{2}y^{2}}\left(\begin{array}{c} \left(x+\gamma x^{2}+\gamma y^{2}\right)\cos\left(\kappa \theta \left(x,y\right)\right)-y\sin\left(\kappa \theta \left(x,y\right)\right)\\ \;+i\left\{\left(x+\gamma x^{2}+\gamma y^{2}\right)\sin\left(\kappa \theta \left(x,y\right)\right)+y\cos\left(\kappa \theta \left(x,y\right)\right)\right\} \end{array}\right).$$
If κ=∞, then we regard κθ(x,y) as *y* because we have $\lim_{\kappa \to \infty }\exp_{\kappa }\left(z\right)=e^{z}=e^{x}\left(\cos y+i\sin y\right)$. Then, $f_{\kappa ,\gamma }\left(z\right)\in R$ implies
(3)$$\left(x+\gamma x^{2}+\gamma y^{2}\right)\sin\left(\kappa \theta \left(x,y\right)\right)+y\cos\left(\kappa \theta \left(x,y\right)\right)=0.$$
If sin(κθ(x,y))=0, then cos(κθ(x,y)) does not vanish so that *y* needs to be zero. This means that if $z=x+yi\in f_{\kappa ,\gamma }\left(R\right)$ with y≠0, then we have sin(κθ(x,y))≠0. Thus, Equation (Equation 3) for y≠0 can be rewritten as
(4)$$F\left(x,y\right):=\left(x+\gamma x^{2}+\gamma y^{2}\right)+y\cot\left(\kappa \theta \left(x,y\right)\right)=0.$$For y=0, we set $F\left(x,0\right):=\lim_{y\to 0}F\left(x,y\right)$. If x=0, then since $\theta \left(0,y\right)=\mathrm{Arctan}\left(\frac{y}{\kappa }\right)$, we have $F\left(0,0\right)=\lim_{y\to 0}y\cot\left(\kappa \mathrm{Arctan}\left(\frac{y}{\kappa }\right)\right)=1$. Here, we take the main branch of Arctanx, i.e., we have Arctan0=0. Let us introduce a connected domain $\Omega =\Omega_{\kappa ,\gamma }$ by
(5)$$\Omega =\Omega_{\kappa ,\gamma }:={\left\{z=x+yi\in D\left(f_{\kappa ,\gamma }\right);\;F\left(x,y\right)>0\right\}}^{\circ },$$
where $A^{\circ }$ denotes the connected component of an open set A⊂C containing z=0. Please note that since *F* is an even function in *y*, the domain Ω is symmetric with respect to the real axis. Set
(6)$$S:=R\backslash f_{\kappa ,\gamma }\left(\Omega_{\kappa ,\gamma }\cap R\right).$$

**Definition** **1.**
*If there exists a unique holomorphic extension $W_{\kappa ,\gamma }$ of $w_{\kappa ,\gamma }$ to $C\backslash \bar{S}$, then we call $W_{\kappa ,\gamma }$ the Lambert–Tsallis function.*


**Remark** **2.**
*Strictly speaking, the Lambert–Tsallis function $W_{\kappa ,\gamma }$ is the main branch of the multivalued Lambert–Tsallis W function (recall that $W_{\kappa ,\gamma }\left(0\right)=0$). In our terminology the Lambert–Tsallis W function is multivalued and the Lambert–Tsallis function $W_{\kappa ,\gamma }$ is single-valued. In this paper, we only study $W_{\kappa ,\gamma }$ among other branches of the Lambert–Tsallis W function.*


Our goal now is to prove the following theorem which allows the determination of the parameter domain of $\left\{S_{\kappa ,\gamma }\right\}$ such that $S_{\kappa ,\gamma }$ is a Stieltjes transformation of a probability measure.

**Theorem** **3.***There exists the main branch $W_{\kappa ,\gamma }$ of the Lambert–Tsallis W function if and only if* (i)*0<|κ|<1 and $\gamma \leq \min(0,\frac{1}{\kappa })$, or*(ii)*|κ|≥1 and $\gamma \leq \frac{1}{4}{\left(1+\frac{1}{\kappa }\right)}^{2}$, or*(iii) *κ=∞ and $\gamma \leq \frac{1}{4}$. Moreover, $W_{\kappa ,\gamma }$ maps $C^{+}$ onto $\Omega^{+}:=\Omega \cap C^{+}$ bijectively.*

We conclude by giving the range of parameters κ,γ such that the corresponding function $S_{\kappa ,\gamma }\left(z\right)$ is the Stieltjes transformation of a probabilistic measure.

**Corollary** **2.**
*$S_{\kappa ,\gamma }\left(z\right)$ is the Stieltjes transformation of a probabilistic measure if and only if:*
(i) *0<|κ|<1 and $\gamma \leq \min(0,\frac{1}{\kappa })$, or* (ii) *|κ|≥1 and $\gamma \leq \frac{1}{4}{\left(1+\frac{1}{\kappa }\right)}^{2}$, or* (iii) *κ=∞ and $\gamma \leq \frac{1}{4}$.*

Recall from [1] that if (i) $\gamma \leq \frac{1}{\kappa }\leq 1$ and γ<1 or (ii) κ=∞ and γ≤0, then there exists a large random matrix model of the form (Equation 2) such that the Stieltjes transformation of its spectral limit is $S_{\kappa ,\gamma }\left(z\right)$. It is an interesting open problem whether or not a large random matrix model exists with $S_{\kappa ,\gamma }\left(z\right)$ as the spectral limit for the following cases:(a)0<κ<1 and γ<0,(b)κ>1 and $\frac{1}{\kappa }<\gamma <\frac{1}{4}{\left(1+\frac{1}{\kappa }\right)}^{2}$,(c)κ=∞ and $0<\gamma \leq \frac{1}{4}$.

Figure 2 indicates the parameter domain on κ and γ. The grey area corresponds to random matrices of the form (Equation 2) and the red area corresponds to Stieltjes transformations of a probabilistic measure in cases (a), (b), (c).

**Remark** **3.**
*As we shall see in Section 5.4, in the case κ>1 and $\gamma >\frac{1}{4}{\left(1+\frac{1}{\kappa }\right)}^{2}$, the function $f_{\kappa ,\gamma }$ maps $\Omega \;\mathrm{to}\;C\backslash \bar{S}$ two-to-one, and hence a holomorphic extension of $w_{\kappa ,\gamma }$ exists on a smaller domain in
Ω
which is mapped by
$f_{\kappa ,\gamma }$ bijectively to $C\backslash \bar{S}$, but it is not unique. In the case 0<κ<1 and γ>0, the function $f_{\kappa ,\gamma }:D\left(f_{\kappa ,\gamma }\right)\to C\backslash \bar{S}$ is not surjective. It is well known that a function S(z) on C is a Stieltjes transformation of a probability measure if and only if S(z) is analytic function of $C^{+}\to C^{+}$ with ${lim sup}_{y\to +\infty }y\left|S\left(iy\right)\right|=1$. Thus, the Theorem 3 shows that $S_{\kappa ,\gamma }$ for parameters outside of the theorem cannot be a Stieltjes transformation of a probability measure.*


**Remark** **4.**
*In [1], we have proven Theorem 3 for the case $\gamma \leq \frac{1}{\kappa }\leq 1$ and γ<1, and when κ=∞ and γ≤0. Please note that the case κ<0 can be derived from the case κ>0; see Section 5.4.5. Therefore, the sufficiency in the Theorem 3 is essentially new in the three cases (a),(b),(c).*

*Please note that $S_{\kappa ,\gamma }\left(z\right)$ for these cases are not obtained from Wishart-type random matrices as in (Equation 2). The necessity part in the Theorem 3 is also a new result.*


## 5. Proof of Theorem 3

The proof of the Theorem 3 is conducted in two steps. More precisely, we first give an explicit expression of $\Omega =\Omega_{\kappa ,\gamma }$, and then show that $f_{\kappa ,\gamma }$ maps Ω to $C\backslash \bar{S}$ bijectively.

### 5.1. Properties of the Generalized Tsallis Function $f_{\kappa ,\gamma }$

In this section, we investigate the function $f_{\kappa ,\gamma }$ in detail to prepare the proof of the Theorem 3. Let us change variables in (Equation 4) by
(7)$$re^{i\theta }=1+\frac{z}{\kappa }\;\left(r>0,\;\theta \in \left(0,\pi \right)\right),\;\mathrm{or}\;\mathrm{equivalently}\;\left\{\begin{array}{c} x=\kappa (r\cos\theta -1),\\ y=\kappa r\sin\theta . \end{array}\right.$$
Set a:=γκ and
(8)$$b\left(\theta \right)=\left(1-2a\right)\cos\theta +\sin\theta \cot\left(\kappa \theta \right)\;\left(\theta \neq 0\right),\;b\left(0\right)=1-2a+\frac{1}{\kappa }.$$
Then, Equation (Equation 4) can be written as
(9)$$ar^{2}+b\left(\theta \right)r+a-1=0.$$
If sin(κθ)≠0, then (Equation 9) has a solution
$$r=r_{\pm }\left(\theta \right)=\left(-b\left(\theta \right)\pm \sqrt{b{\left(\theta \right)}^{2}-4a\left(a-1\right)}\right)/\left(2a\right).$$
Thus, for each angle θ, there exists at most two points on $f_{\kappa ,\gamma }^{-1}\left(R\right)$. Since the change (Equation 7) of variables is the polar transformation, we need to know whether $r_{\pm }\left(\theta \right)$ is positively real or not. To do so, we shall study the function
$$D\left(\theta \right):=b{\left(\theta \right)}^{2}-4a\left(a-1\right).$$

**Remark** **5.**
*Intuitively, if D(θ)<0 for some $\theta <\frac{\pi }{\kappa }$ then the domain $\Omega_{\kappa ,\gamma }$ become bigger and the function $f_{\kappa ,\gamma }$ maps $\Omega_{\kappa ,\gamma }\cap R$ to the whole R, which threatens the unicity of $W_{\kappa ,\gamma }$. On the other hand, if $\sin\left(\kappa \theta (x,y)\right)=0$ then F(x,y)<0 which implies that the angle θ such that F(x,y)=0 is included in $\left(0,\frac{\pi }{\kappa }\right)$. When $\frac{\pi }{\kappa }>\pi$, or equivalently κ<1, it may fail that F(x,y)=0 for $x+yi\in D(f_{\kappa ,\gamma })$, which violates the bijectivity of $f_{\kappa ,\gamma }$. The discussions below will justify such intuitive arguments with a slightly complicated calculation.*


We first consider $b^{\prime }\left(\theta \right)$ because $D^{\prime }\left(\theta \right)=2b\left(\theta \right)b^{\prime }\left(\theta \right)$. Set
$$G\left(\kappa \right):=\frac{2\kappa^{2}+3\kappa +1}{6\kappa }.$$
Since b(θ)=2(a−1)sinθ for $\kappa =1,\frac{1}{2}$, we exclude these two cases.

**Lemma** **1.**
*Let us set $I_{0}:=\left(0,\min\left(\pi ,\frac{\pi }{\kappa }\right)\right)$.*



(1)
*Suppose that $0<\kappa <\frac{1}{2}$, or κ>1. If a>G(κ), then there exists a unique $\varphi_{*}\in I_{0}$ such that $b^{\prime }\left(\varphi_{*}\right)=0$, and one has $b^{\prime }\left(\theta \right)>0$ for $\theta \in (0,\varphi_{*})$, and $b^{\prime }\left(\theta \right)<0$ otherwise. If a<G(κ), then one has $b^{\prime }\left(\theta \right)<0$ for any $\theta \in I_{0}$.*
(2)
*Suppose that $\frac{1}{2}<\kappa <1$. If a≤G(κ), then there exists a unique $\varphi_{*}\in I_{0}$ such that $b^{\prime }\left(\varphi_{*}\right)=0$, and one has $b^{\prime }\left(\theta \right)<0$ for $\theta \in (0,\varphi_{*})$, and $b^{\prime }\left(\theta \right)>0$ for $\theta \in (\varphi_{*},\pi )$. If a≥G(κ), then one has $b^{\prime }\left(\theta \right)>0$ for any $\theta \in I_{0}$.*



**Proof.** Let us set for α,κ∈R
(10)$$\left\{\begin{array}{c} H_{\alpha }\left(x\right):=\sin\left(\alpha x\right)-\alpha \sin\left(x\right),\\ F_{\kappa }\left(x\right):=\tan x\cdot \cot\left(\kappa x\right) \end{array}\right.\;J_{\kappa }\left(x\right):=-\frac{2x-2\kappa -1}{4\kappa x-2\kappa -1}.$$

For $\kappa \neq 1,\frac{1}{2}$, we have
$$b\left(\theta \right)=\cos\theta +\sin\theta \cot\left(\kappa \theta \right)-2a\cos\theta =\frac{\cos\theta \sin(\kappa \theta )+\sin\theta \cos(\kappa \theta )}{\sin(\kappa \theta )}-2a\cos\theta ,$$
and hence $b^{\prime }\left(\theta \right)$ can be written as
$$b^{\prime }\left(\theta \right)=\frac{H_{2\kappa +1}\left(\theta \right)}{2\sin^{2}\left(\kappa \theta \right)}+2a\sin\theta .$$Let us set
(11)$$\left\{\begin{array}{c} B\left(\theta \right):=2\sin^{2}\left(\kappa \theta \right)b^{\prime }\left(\theta \right)=H_{2\kappa +1}\left(\theta \right)+4a\sin\theta \sin^{2}\left(\kappa \theta \right),\\ \ell (\kappa ,a):=4a\kappa -2\kappa -1. \end{array}\right.$$Then, the signatures of $b^{\prime }\left(\theta \right)$ and B(θ) are the same on the interval $I_{0}$, and hence we actually work with B(θ). We have B(0)=0 for any κ>0, and
(12)$$B\left(\pi \right)=-\cot\left(\kappa \pi \right)\sin^{2}\left(\kappa \pi \right)\left\{\begin{array}{cc} <0 & (\mathrm{if}\;0<\kappa <\frac{1}{2}),\\ >0 & \left(\mathrm{if}\;\frac{1}{2}<\kappa <1\right) \end{array}\right.$$
and
$$B\left(\frac{\pi }{\kappa }\right)=-2\kappa \sin\frac{\pi }{\kappa }<0\;\left(\kappa >1\right).$$In fact, we have $B\left(\frac{\pi }{\kappa }\right)=H_{2\kappa +1}\left(\frac{\pi }{\kappa }\right)=-2\kappa \sin\frac{\pi }{\kappa }<0$ when κ>1. The derivative of B(θ) is given as
(13)$$B^{\prime }\left(\theta \right)=\left\{\begin{array}{cc} 2\ell \left(\kappa ,a\right)\cos\theta \sin^{2}\left(\kappa \theta \right)\left(F_{\kappa }\left(\theta \right)-J_{\kappa }\left(a\right)\right)\; & (\ell (\kappa ,a)\neq 0),\\ \frac{1-4\kappa^{2}}{\kappa }\cos\theta \sin^{2}\left(\kappa \theta \right)\; & (\ell (\kappa ,a)=0). \end{array}\right.$$We note that the condition G(κ)=a comes from a condition $F_{\kappa }\left(0\right)-J_{\kappa }\left(a\right)=0$. According to a case classification with respect to $F_{\kappa }\left(\theta \right)$ (cf. Table 1), we need to consider the following four cases: (i) $0<\kappa <\frac{1}{2}$, (ii) $\frac{1}{2}<\kappa <1$, (iii) 1<κ<2, and (iv) κ≥2.

Let us first consider the equation
(14)$$F_{\kappa }\left(\theta \right)-J_{\kappa }\left(a\right)=0\Leftrightarrow F_{\kappa }\left(\theta \right)=J_{\kappa }\left(a\right)\;\mathrm{on}\;I_{0}=\left(0,\min\left(\pi ,\frac{\pi }{\kappa }\right)\right).$$Since $J_{\kappa }\left(a\right)$ does not depend on θ, it is important to know where it meets end points or maximal/minimal values of $F_{\kappa }\left(\theta \right)$. By Table 1, for end points, they are $J_{\kappa }\left(a\right)=\frac{1}{\kappa }$ or 0, and its solutions on *a* are given as a=G(κ) or a=κ+1/2, respectively. If $\frac{1}{2}<\kappa <2$ (κ≠1), then we need a solution of $J_{\kappa }\left(a\right)=F_{\kappa }\left(x_{*}\right)$. Since $J_{\kappa }$ is bijective, there exists a unique solution $a_{*}$ such that $J_{\kappa }\left(a_{*}\right)=F_{\kappa }\left(x_{*}\right)$.

The proof can be done by considering further cases according to the position of $J_{\kappa }\left(a\right)$. The calculations are delicate and tedious but elementary, and, thus, we only give proof for the case $\frac{1}{2}<\kappa <1$ and $0\leq J_{\kappa }\left(a\right)\leq F_{\kappa }\left(x_{*}\right)$.

Assume that $\frac{1}{2}<\kappa <1$. In this case, we have B(π)>0 by (Equation 12). Recall that $J_{\kappa }$ is monotonic decreasing in *a* in this situation. Let $x_{*}$ be the maximal point of $F_{\kappa }$ in $I_{0}$. Then, we have $0<F_{\kappa }\left(x_{*}\right)<1$. In fact, since $x_{*}\in \left(\frac{\pi }{2},\pi \right)$, we have $\sin(\left(1-\kappa \right)x_{*})>0$ and hence it implies that
$$\sin x_{*}\cos\left(\kappa x_{*}\right)-\cos x_{*}\sin\left(\kappa x_{*}\right)>0\Leftrightarrow F_{\kappa }\left(x_{*}\right)=\frac{\sin x_{*}\cos\left(\kappa x_{*}\right)}{\cos x_{*}\sin\left(\kappa x_{*}\right)}<1.$$Here we use $\cos x_{*}<0$ and $\sin(\kappa x_{*})>0$. Let $a_{*}$ be the unique solution of $F_{\kappa }\left(x_{*}\right)-J_{\kappa }\left(a\right)=0$ in *a*. Then, we see that $a_{*}\in \left(1,\frac{1}{2}+\kappa \right)$ by $J_{\kappa }\left(1\right)=1$. Table 1 tells us that we have three cases in Equation (Equation 14) as follows.

(a)It does not have a solution if $F_{\kappa }\left(x_{*}\right)<J_{\kappa }\left(a\right)<\frac{1}{\kappa }$, which is equivalent to the condition $G\left(\kappa \right)<a<a_{*}$. Since $F_{\kappa }\left(x_{*}\right)>0$, we see that ℓ(κ,a)>0 in this situation.(b)The Equation (Equation 14) has a unique solution φ if $J_{\kappa }\left(a\right)\geq \frac{1}{\kappa }$ or $J_{\kappa }\left(a\right)<0$. In the former case, the condition is equivalent to $\frac{1}{2}+\frac{1}{4\kappa }<a\leq G\left(\kappa \right)$, and we have $\varphi <\frac{\pi }{2}$ and ℓ(κ,a)>0. In the latter case, we have $\varphi >\frac{\pi }{2}$, and the condition is divided into two situations; one is $a>\frac{1}{2}+\kappa$, and the other is $a<\frac{1}{2}+\frac{1}{4\kappa }$.(c)The Equation (Equation 14) has two solutions $\varphi_{1}\leq \varphi_{2}$ if $0\leq J_{\kappa }\left(a\right)\leq F_{\kappa }\left(x_{*}\right)$, which is equivalent to the condition $a_{*}\leq a\leq \frac{1}{2}+\kappa$ so that ℓ(κ,a)>0. We note that we have $\varphi_{i}>\frac{\pi }{2}$, and if $J_{\kappa }\left(a\right)=F_{\kappa }\left(x_{*}\right)$ then we have $\varphi_{1}=\varphi_{2}$. We only deal with case (c).

Let us assume that $0\leq J_{\kappa }\left(a\right)\leq F_{\kappa }\left(x_{*}\right)$, i.e., $a_{*}\leq a\leq \frac{1}{2}+\kappa$. In this situation, the signature of $B^{\prime }$ is negative in the interval $\left(0,\varphi_{1}\right)$ and $\left(\varphi_{2},\pi \right)$, and positive in $\left(\varphi_{1},\varphi_{2}\right)$. To show that $B(\varphi_{i})>0$ (i=1,2), which implies $b^{\prime }\left(\theta \right)>0$ for any $\theta \in I_{0}$, we calculate $b^{\prime }\left(\theta \right)$ in a different way as follows. By differentiating b(θ) using expression (Equation 8), we obtain
$$b^{\prime }\left(\theta \right)=\left(2a-1\right)\sin\theta +\cos\theta \cot\left(\kappa \theta \right)-\frac{\kappa \sin\theta }{\sin^{2}\left(\kappa \theta \right)},$$
and hence B(θ) can be also described as
(15)$$B\left(\theta \right)=2\left(2a-\kappa -1\right)\sin\theta +2\cos\theta \sin\left(\kappa \theta \right)\cos\left(\kappa \theta \right)(\left(2a-1\right)F_{\kappa }\left(\theta \right)+1).$$Since $\varphi_{i}$(i=1,2) satisfy $F_{\kappa }\left(\varphi_{i}\right)-J_{\kappa }\left(a\right)=0$ by definition, we see that
(16)$$\begin{array}{ccc} \left(2a-1\right)F_{\kappa }\left(\varphi_{i}\right)+1\; & = & \left(2a-1\right)J_{\kappa }\left(a\right)+1\\ \; & = & \frac{-(2a-1)(2a-2\kappa -1)+(4a\kappa -2\kappa -1)}{\ell (\kappa ,a)}\\ \; & = & \frac{4a(a-1)}{\ell (\kappa ,a)}. \end{array}$$Now we assume $a_{*}<a<\frac{1}{2}+\kappa$ and we know $a_{*}>1$ so that we have 2a−κ−1>0 and $\left(2a-1\right)F_{\kappa }\left(\varphi_{i}\right)+1>0$ for i=1,2 by (Equation 16). Since $\varphi_{i}\in \left(\frac{\pi }{2\kappa },\pi \right)$, we see that $\cos\varphi_{i}\sin\left(\kappa \varphi_{i}\right)$$\cos(\kappa \varphi_{i})>0$ and hence we obtain $B(\varphi_{i})>0$(i=1,2) by (Equation 15). Thus, we conclude $b^{\prime }\left(\theta \right)>0$ on the interval $I_{0}$.

The other cases can be done similarly. □

### 5.2. The domain Ω for 0<κ<+∞

Now, we can obtain explicit formulas for Ω. Please note that since F(x,y) is a continuous function, the boundary ∂Ω is included in the set $\left\{z=x+yi\in C;\;F(x,y)=0\right\}\subset f_{\kappa ,\gamma }^{-1}\left(R\right)$.

**Proposition** **1.**
*Suppose that κ=1.*



(1)
*If γ>1, then $\Omega =C\backslash \{-\frac{1}{\gamma }\}$.*
(2)
*If 0<γ≤1, then $\Omega =\left\{z=x+yi\in D\left(f_{\kappa ,\gamma }\right);\;{\left(x+\frac{1}{\gamma }\right)}^{2}+y^{2}>\frac{1-\gamma }{\gamma^{2}}\right\}$.*
(3)
*If γ=0, then $\Omega =\left\{z=x+yi\in D(f_{\kappa ,\gamma });\;1+2x>0\right\}$.*
(4)
*If γ<0, then $\Omega =\left\{z=x+yi\in D\left(f_{\kappa ,\gamma }\right);\;{\left(x+\frac{1}{\gamma }\right)}^{2}+y^{2}<\frac{1-\gamma }{\gamma^{2}}\right\}$, which is bounded.*



**Proof.** In the case κ=1, we have θ(x,y)=Arg(1+z) so that tan(θ(x,y))=y/(1+x). Thus, we have
$$F\left(x,y\right)=1+2x+\gamma x^{2}+\gamma y^{2}=\left\{\begin{array}{cc} \gamma {\left(x+\frac{1}{\gamma }\right)}^{2}+\gamma y^{2}+\frac{\gamma -1}{\gamma } & (\mathrm{if}\;\gamma \neq 0),\\ 1+2x & (\mathrm{if}\;\gamma =0). \end{array}\right.$$Please note that z=0 is contained in the set
$$\left\{z=x+yi\in C;\;1+2x+\gamma x^{2}+\gamma y^{2}>0\right\}.$$Since it is connected, the proof is completed. □

Set $\theta_{0}:=\frac{\pi }{\kappa }$ and $I_{0}=\left(0,\min\left(\pi ,\theta_{0}\right)\right)$. Let $r_{\pm }\left(\theta \right)$ be the solutions of Equation (Equation 9) and let $\alpha_{i}$, i=1,2 be the solutions of the equation $f_{\kappa ,\gamma }\left(z\right)=0$ which come from rational part of $f_{\kappa ,\gamma }$. If $\alpha_{i}$ are real, then we assume that $\alpha_{1}\leq \alpha_{2}$, and if not, then we assume that Imα1>Imα2. Recall that a=κγ and $D\left(0\right)={\left(1+\frac{1}{\kappa }\right)}^{2}-4\gamma$.

**Proposition** **2.**
*Let κ>0 with κ≠1. For $z\in D(f_{\kappa ,\gamma })$, one sets $re^{i\theta }=1+\frac{z}{\kappa }$. Then, Ω can be described as follows.*



(1)
*If a<0, then there exists a unique $\theta_{*}\in \left(0,\frac{\pi }{\kappa +1}\right)$ such that $r_{+}\left(\theta_{*}\right)=r_{-}\left(\theta_{*}\right)$ and that $r_{\pm }\left(\theta \right)$ are both positive with $r_{-}\left(\theta \right)\leq r_{+}\left(\theta \right)$ on $\left(0,\theta_{*}\right)$. Moreover,*

$$\Omega =\left\{z\in D\left(f_{\kappa ,\gamma }\right);\;\left|\theta \right|<\theta_{*}\;\mathrm{and}\;r_{-}\left(\theta \right)<r<r_{+}\left(\theta \right)\right\}.$$


*In particular, Ω is bounded. One has $\alpha_{1},\alpha_{2}\in \partial \Omega$ and $-\frac{1}{\gamma }\in \bar{\Omega }$, whereas $-\kappa \notin \bar{\Omega }$.*
(2)
*If a=0, then one has $r\left(\theta \right)=r_{\pm }\left(\theta \right)=\frac{\sin(\kappa \theta )}{\sin((\kappa +1)\theta )}$ which is positive on the interval $\left(0,\frac{\pi }{\kappa +1}\right)$. Moreover,*

$$\Omega =\left\{z\in D\left(f_{\kappa ,\gamma }\right);\;\left|\theta \right|<\frac{\pi }{\kappa +1}\;\mathrm{and}\;r>r\left(\theta \right)\right\}.$$



Ω

* has an asymptotic line $y=\pm \left(\tan\frac{\pi }{\kappa +1}\right)\left(x+\frac{\kappa^{2}}{\kappa +1}\right)$. One has $\alpha_{1}=\alpha_{2}=-\frac{\kappa }{\kappa +1}\in \partial \Omega$ and $-\kappa \notin \bar{\Omega }$.*
(3)
*If 0<a<1, then $r_{+}\left(\theta \right)$ is the only positive solution of (9) on $I_{0}$, and*

$$\Omega =\left\{z\in D\left(f_{\kappa ,\gamma }\right);\;\left|\theta \right|<\min\left(\theta_{0},\pi \right)\;\mathrm{and}\;r>r_{+}\left(\theta \right)\right\}.$$


*One has $\alpha_{2}\in \partial \Omega$, whereas $\alpha_{1},-\frac{1}{\gamma },-\kappa \notin \bar{\Omega }$. If κ>1, then Ω has an asymptotic line $y=\pm \left(\tan\theta_{0}\right)\left(x+\kappa -\frac{1}{a}\right)$. Please note that if 0<κ<1 then $\Omega =D(f_{\kappa ,\gamma })$.*
(4)
*Suppose that a=1 and κ≠1.*
(a)
*If 0<κ<1, then one has $r_{+}\left(\theta \right)=0$ and $r_{-}\left(\theta \right)=-b\left(\theta \right)<0$ for $\theta \in I_{0}$, and $\Omega =D(f_{\kappa ,\gamma })$. One has $\alpha_{1}=-1\in \partial \Omega$ and $\alpha_{2}=-\kappa =-\frac{1}{\gamma }\in \partial \Omega$.*
(b)
*If κ>1, then one has $r_{+}\left(\theta \right)=-b\left(\theta \right)>0$ and $r_{-}\left(\theta \right)=0$ for $\theta \in I_{0}$, and*

Ω=z∈D(fκ,γ);|θ|<θ0andr>r+(θ).


*One has $\alpha_{2}=-1\in \partial \Omega$, while $\alpha_{1}=-\kappa =-\frac{1}{\gamma }\notin \bar{\Omega }$. Moreover, Ω has an asymptotic line $y=\pm \left(\tan\theta_{0}\right)\left(x+\kappa -\frac{1}{a}\right)$.*
(5)
*Suppose that a>1.*
(a)
*If κ>1 with D(0)≥0, then $r_{\pm }\left(\theta \right)$ are both positive in $I_{0}$ with $r_{-}\left(\theta \right)\leq r_{+}\left(\theta \right)$, and*

$$\Omega =\left\{z\in D\left(f_{\kappa ,\gamma }\right);\;\left|\theta \right|<\theta_{0}\;\mathrm{and}\;r>r_{+}\left(\theta \right)\right\}.$$



Ω

* has an asymptotic line $y=\pm \left(\tan\theta_{0}\right)\left(x+\kappa -\frac{1}{a}\right)$. One has $\alpha_{2}\in \partial \Omega$, but $-\kappa ,-\frac{1}{\gamma }$, $\alpha_{1}\notin \bar{\Omega }$.*
(b)
*If κ>1 and D(0)<0, then there exists a unique $\theta_{*}\in \left(0,\theta_{0}\right)$ such that $D(\theta_{*})=0$, and $r_{\pm }\left(\theta \right)$ are both positive in the interval $\left(\theta_{*},\theta_{0}\right)$. Moreover,*

$$\Omega =\left\{z\in D\left(f_{\kappa ,\gamma }\right);\;\begin{array}{c} |\theta |<\theta_{0}\;\mathrm{and}\;\mathrm{if}\;\left|\theta \right|\geq \theta_{*}\;\mathrm{then}\;\\ 0<r<r_{-}\left(\theta \right)\;\mathrm{or}\;r>r_{+}\left(\theta \right) \end{array}\right\}.$$


*In this case, $\alpha_{i}$, i=1,2 are both non-real numbers and one has $\alpha_{i},-\kappa \in \partial \Omega$ and $-\frac{1}{\gamma }\in \bar{\Omega }$. Moreover, Ω has an asymptotic line $y=\pm \left(\tan\theta_{0}\right)\left(x+\kappa -\frac{1}{a}\right)$.*
(c)
*If 0<κ<1, then there is no $\theta \in I_{0}$ such that D(θ)>0, and one has $\Omega =D(f_{\kappa ,\gamma })$.*



**Proof.** Since Ω is symmetric with respect to the real axis, we shall actually work with the boundary $\partial \Omega^{+}$ of $\Omega^{+}=\Omega \cap C^{+}$. The cases (1)–(3) are done in the previous paper [1], and thus we omit them.(4) Assume that a=1 and κ≠1. In this case, $\alpha_{i}$, i=1,2 are given as $\alpha_{i}=-1$ or $-\frac{1}{\gamma }=-\kappa$, and Equation (Equation 9) reduces to $r^{2}+b\left(\theta \right)r=0$, whose solutions are r(θ)=0, −b(θ). Since b(θ) can be described as b(θ)=sin((1−κ)θ)/sin(κθ) in this case, it is easily verified that, if 0<κ<1 then b(θ)>0, and if κ>1 then b(θ)<0 for any $\theta \in I_{0}$. This means that if 0<κ<1 then the Equation (Equation 9) does not have a positive solution, and hence we have
$$\Omega^{+}=\left\{z\in D(f_{\kappa ,\gamma });\;0<\theta <\pi ,\;r>0\right\}=C^{+},$$
which shows $\Omega =D(f_{\kappa ,\gamma })$. Since we have $\alpha_{1}=-1$ and $\alpha_{2}=-\kappa =-\frac{1}{\gamma }$, the assertion (4)-(a) is proved. On the other hand, if κ>1, then we have $b^{\prime }\left(\theta \right)<0$ for $\theta \in I_{0}$ by Lemma 1 together with the fact G(κ)>1 for any κ>1. Since $b\left(0\right)=1+\frac{1}{\kappa }-2<0$ and $\lim_{\theta \to \theta_{0}-0}b\left(\theta \right)=-\infty$, the function D(θ) is monotonic increasing on $I_{0}$ and therefore we have $\lim_{\theta \to \theta_{0}-0}r_{+}\left(\theta \right)=+\infty$. Thus, the curve $r_{+}\left(\theta \right)$, $\theta \in I_{0}$ has an asymptotic line with gradient $\tan\theta_{0}$, which is determined later. Therefore, we have
$$\Omega^{+}=\left\{z\in D\left(f_{\kappa ,\gamma }\right);\;0<\theta <\theta_{0}\;\mathrm{and}\;r>r_{+}\left(\theta \right)\right\}.$$In this case, we have $\alpha_{1}=-\kappa =-\frac{1}{\gamma }$ and $\alpha_{2}=-1$. The assertion (4)-(b) is now proved.(5) Suppose that a>1 and κ≠1. In this case, we have $b\left(0\right)=1+\frac{1}{\kappa }-2a$ and
$$\lim_{\theta \to \theta_{0}-0}b\left(\theta \right)=-\infty \;\left(\mathrm{if}\;\kappa >1\right),\;b\left(\pi \right)=2a-1>0\;\left(0<\kappa <1\right).$$Please note that b(0)>0 if κ>1. Since $r_{+}\left(\theta \right)\cdot r_{-}\left(\theta \right)=\frac{a-1}{a}>0$, two solutions $r_{\pm }\left(\theta \right)$ of (Equation 9) have the same signature if $r_{\pm }\left(\theta \right)$ are real.(a) We first consider the case κ>1 and D(0)≥0. Let us show that D(θ)>0 for $\theta \in I_{0}$. If we set $K\left(x\right):=\frac{x}{4}{\left(1+\frac{1}{x}\right)}^{2}$, then the condition D(0)≥0 is equivalent to a≤K(κ), and hence we have a≤G(κ) because $G\left(x\right)-K\left(x\right)=\left(x^{2}-1\right)/\left(12x\right)>0$ if x>1. By the assumption κ>1, we see that $b^{\prime }\left(\theta \right)<0$ for any $\theta \in I_{0}$ by Lemma 1 so that *b* is monotonically decreasing in this interval. Since b(0)<0, the function *b* is negative in $I_{0}$, and hence $D^{\prime }\left(\theta \right)=2b\left(\theta \right)b^{\prime }\left(\theta \right)>0$ so that D(θ) is monotonic increasing in the interval $I_{0}$, and in particular, it is positive on $I_{0}$. Thus, we see that $r_{\pm }\left(\theta \right)$ are real on $I_{0}$, and by $r_{+}\left(\theta \right)+r_{-}\left(\theta \right)=-b\left(\theta \right)/a>0$ we have $r_{\pm }\left(\theta \right)>0$$\left(\theta \in I_{0}\right)$. Since for ε=±1
(17)$$r_{\varepsilon }^{\prime }\left(\theta \right)=\frac{1}{2a}\left(-b^{\prime }\left(\theta \right)+\varepsilon \frac{2b\left(\theta \right)b^{\prime }\left(\theta \right)}{2\sqrt{D(\theta )}}\right)=-\varepsilon b^{\prime }\left(\theta \right)\frac{r_{\varepsilon }\left(\theta \right)}{\sqrt{D(\theta )}},$$
we have $r_{+}^{\prime }\left(\theta \right)>0$, and hence the function $r_{+}\left(\theta \right)$ is monotonic increasing, whereas $r_{-}\left(\theta \right)$ is monotonic decreasing because $r_{-}\left(\theta \right)=\frac{a-1}{ar_{+}\left(\theta \right)}$. Please note that we have $r_{+}\left(\theta \right)\to +\infty$ and $r_{0}\left(\theta \right)\to 0$ as $\theta \to \theta_{0}-0$. Thus, $r_{+}\left(\theta \right)$, $\theta \in I_{0}$ draws an unbounded curve connecting $z=\alpha_{2}$ to *∞* with an asymptotic line with slope $\tan\theta_{0}$, and $r_{-}\left(\theta \right)$, $\theta \in I_{0}$ draws a bounded curve connecting $z=\alpha_{1}$ to z=−κ. Since we have $-\kappa <-\frac{1}{\gamma }<\alpha_{1}<\alpha_{2}<0$ and since Ω is the connected component including z=0, we have
$$\Omega^{+}=\left\{z\in D\left(f_{\kappa ,\gamma }\right);\;0<\theta <\theta_{0},\;r>r_{+}\left(\theta \right)\right\}.$$(b) Next, we assume that κ>1 and D(0)<0. According to Lemma 1 (1), we consider the function $b^{\prime }\left(\theta \right)$ in two cases, that is, (i) a≤G(κ) and (ii) a>G(κ).(i) Assume that a≤G(κ). Then, $b^{\prime }$ is negative and hence b(θ) is monotonically decreasing. Since b(0)<0, we see that b(θ)<0 for any $\theta \in I_{0}$ and therefore $D^{\prime }\left(\theta \right)=2b\left(\theta \right)b^{\prime }\left(\theta \right)>0$ for any $\theta \in I_{0}$. Thus, D(θ) is monotonic increasing with D(0)<0. Since D(θ)→+∞ as $\theta \to \theta_{0}-0$, we see that there exists a unique $\theta_{*}$ such that $D(\theta_{*})=0$, and $r_{\pm }\left(\theta \right)$ are real for $\theta \in (\theta_{*},\theta_{0})$. In this interval, since $r_{+}\left(\theta \right)+r_{-}\left(\theta \right)=-b\left(\theta \right)/a>0$, we see that $r_{\pm }\left(\theta \right)$ are both positive. By (Equation 17), we have $r_{+}^{\prime }\left(\theta \right)>0$ and thus the function $r_{+}\left(\theta \right)$ is monotonic increasing, whereas $r_{-}\left(\theta \right)$ is monotonic decreasing because $r_{-}\left(\theta \right)=\frac{a-1}{ar_{+}\left(\theta \right)}$. As $\theta \to \theta_{0}-0$, we have $r_{+}\left(\theta \right)\to +\infty$ and $r_{-}\left(\theta \right)\to 0$. This means that $r_{+}\left(\theta \right)$ draws an unbounded curve connecting $z=\alpha_{1}$ and z=∞, and $r_{-}\left(\theta \right)$ draws a bounded curve connecting $z=\alpha_{1}$ and z=−κ, where $\alpha_{1}$ is the complex solution of $f_{\kappa ,\gamma }\left(z\right)=0$ with positive imaginary part. Since we have $-\kappa <-\frac{1}{\gamma }<0$, the domain $\Omega^{+}$ is given as
$$\Omega^{+}=\left\{z\in D\left(f_{\kappa ,\gamma }\right);\;\begin{array}{c} 0<\theta <\theta_{0},\;\mathrm{and}\\ \mathrm{if}\;\theta \geq \theta_{*}\;\mathrm{then}\;0<r<r_{-}\left(\theta \right)\;\mathrm{or}\;r>r_{+}\left(\theta \right) \end{array}\right\}.$$(ii) Assume that a>G(κ). In this case, we have D(0)<0 and D(θ)→+∞ as $\theta \to \theta_{0}-0$. Then, there are two possibilities on $b(\varphi_{*})$, i.e., it is positive or negative. If $b(\varphi_{*})\leq 0$, then D(θ) has a unique minimal point at $\theta =\varphi_{*}$. On the other hand, if $b(\varphi_{*})>0$, then there exist exactly two $\varphi_{1}<\varphi_{2}$ in $I_{0}$ such that $b(\varphi_{i})=0$. Then, D(θ) has a unique maximal point at $\theta =\varphi_{*}$ whereas there are exactly two minimal points at $\theta =\varphi_{1},\varphi_{2}$ in $I_{0}$. We note that $D\left(\varphi_{i}\right)=b{\left(\varphi_{i}\right)}^{2}-4a\left(a-1\right)<0$. Since $r_{+}\left(\theta \right)+r_{-}\left(\theta \right)=-b\left(\theta \right)/a$, if $D(\varphi_{*})>0$, then $r_{\pm }\left(\theta \right)$ are both negative so we need not deal with this case. Thus, regardless of whether $b(\varphi_{*})$ is positive or not, there exists a unique $\theta_{*}\in I_{0}$ such that $D(\theta_{*})=0$ and $r_{\pm }\left(\theta \right)>0$ for any $\theta \in (\theta_{*},\theta_{0})$. By (Equation 17), we see that $r_{+}\left(\theta \right)$ is monotonic increasing on $\left(\theta_{*},\theta_{0}\right)$, whereas $r_{-}\left(\theta \right)$ is monotonic decreasing. Moreover, we have $r_{+}\left(\theta \right)\to +\infty$ and $r_{-}\left(\theta \right)\to 0$ as $\theta \to \theta_{0}-0$, and hence the curves $r_{\pm }\left(\theta \right)$, $\theta \in (\theta_{*},\theta_{0})$ form an unbounded curve connecting $z=\alpha_{1}$ and z=∞. Since $-\kappa <-\frac{1}{\gamma }<0$, we have
$$\Omega^{+}=\left\{z\in D\left(f_{\kappa ,\gamma }\right);\;\begin{array}{c} 0<\theta <\theta_{0},\;\mathrm{and}\\ \mathrm{if}\;\theta \geq \theta_{*}\;\mathrm{then}\;0<r<r_{-}\left(\theta \right)\;\mathrm{or}\;r>r_{+}\left(\theta \right) \end{array}\right\}.$$(c) We finally assume that 0<κ<1. In this case, we have $D\left(\pi \right)={\left(2a-1\right)}^{2}-4a\left(a-1\right)=1$. We note that b(0)<0 implies D(0)<0. In fact, b(0)<0 means $1+\frac{1}{\kappa }<2a$ so that
(18)$$D\left(0\right)={\left(1+\frac{1}{\kappa }\right)}^{2}-\frac{4a}{\kappa }<2a\left(1+\frac{1}{\kappa }\right)-\frac{4a}{\kappa }=2a\left(1-\frac{1}{\kappa }\right)<0.$$Since a>1, the signatures of $r_{\pm }\left(\theta \right)$ are the same, and they are the opposite of the signature of b(θ).Let $I_{D}\subset I_{0}$ be the maximal interval such that *D* is positive on $I_{D}$. We shall show that there are no suitable solutions of (Equation 9), i.e., $r_{\pm }\left(\theta \right)<0$ for any $\theta \in I_{D}$, which yields that $\Omega^{+}=C^{+}$ and hence $\Omega =D(f_{\kappa ,\gamma })$. Let us recall Lemma 1.(i) Assume that $0<\kappa <\frac{1}{2}$ and a>G(κ). In this case, b(θ) has a unique maximal point at $\theta =\varphi_{*}$, and we have b(π)>0. If b(0)≥0, then we see that b(θ)>0 for any $\theta \in I_{0}$, which implies $r_{\pm }\left(\theta \right)<0$ for any $\theta \in I_{D}$. If b(0)<0, then there exists a unique $0<\varphi <\varphi_{*}$ such that b(φ)=0, and we have D(0)<0 by (Equation 18). Hence, D(θ) have a unique maximal point $D(\varphi_{*})>0$ at $\theta =\varphi_{*}$ and a unique minimal point D(φ)<0 at θ=φ. Thus, $I_{D}$ is included in (φ,π) and *b* is positive on $I_{D}$, whence $r_{\pm }\left(\theta \right)<0$ for any $\theta \in I_{D}$.(ii) Assume that $0<\kappa <\frac{1}{2}$ and a≤G(κ). Then, Lemma 1 tells us that $b^{\prime }\left(\theta \right)<0$ for any $\theta \in I_{0}$. Thus, b(θ) is monotonic decreasing on the interval $I_{0}$ with b(π)>0, and hence b(θ)>0 for any $\theta \in I_{0}$. This means that $D^{\prime }\left(\theta \right)<0$ and D(θ) is monotonic decreasing on $I_{0}$. Since D(π)=1, we see that $I_{D}=I_{0}$ and hence $r_{\pm }\left(\theta \right)<0$ for any $\theta \in I_{D}$.(iii) Assume that $\frac{1}{2}\leq \kappa <1$. In this case, we have we have $b^{\prime }\left(\theta \right)>0$ for any $\theta \in I_{0}$ so that b(θ) is monotonic increasing. In fact, if $\kappa >\frac{1}{2}$, then since G(κ)<1 for $\frac{1}{2}<\kappa <1$ and since a>1, we always have a>G(κ) so that $b^{\prime }\left(\theta \right)>0$ by Lemma 1, and if $\kappa =\frac{1}{2}$ then we have $b^{\prime }\left(\theta \right)=2\left(a-1\right)\sin\theta$ so that $b^{\prime }\left(\theta \right)>0$. If b(0)≥0, then we have b(θ)≥0 for any $\theta \in I_{0}$ and hence we see that $r_{\pm }\left(\theta \right)<0$ for $\theta \in I_{D}$. If b(0)<0 then there exists a unique φ such that b(φ)=0. Thus, D(θ) has a unique minimal point D(φ)<0 with D(0)<0 and D(π)>0, and hence there are no $\theta \in I_{0}$ such that D(θ)>0 and b(θ)>0. Thus, we have b(θ)>0 for $\theta \in I_{D}$, which implies $r_{\pm }\left(\theta \right)<0$.We shall determine an asymptotic line with respect to $\Omega^{+}$ when $r_{+}\left(\theta \right)\to +\infty$ as $\theta \to \frac{\pi }{\kappa }$ or $\frac{\pi }{\kappa +1}$. To calculate them in one scheme, we set $\vartheta =\frac{\pi }{\kappa }$ or $\frac{\pi }{\kappa +1}$, and denote its denominator by *k*. A line with gradient tanϑ can be written as xsinϑ−ycosϑ=A with some constant *A*. Let us determine the constant *A*. Since x=κ(r(θ)cosθ−1) and y=κr(θ)sinθ as in (Equation 7), we have
$$\begin{array}{ccc} x\sin\vartheta -y\cos\vartheta & = & \kappa \left\{\sin\vartheta (r(\theta )\cos\theta -1)-\cos\vartheta \cdot r(\theta )\sin\theta \right\}\\ & = & -\kappa \left\{r(\theta )\sin(\theta -\vartheta )+\sin\vartheta \right\}. \end{array}$$A simple calculation yields that $\frac{\sin(\theta -\vartheta )}{\sin(k\vartheta )}\to -\frac{1}{k}$ as $\theta \to \theta_{0}-0$, which implies
$$\begin{array}{c} \lim_{\theta \to \vartheta -0}b\left(\theta \right)\sin\left(\theta -\vartheta \right)\\ \;=\lim_{\theta \to \vartheta -0}\left(\left(1-2a\right)\cos\theta \sin\left(\theta -\vartheta \right)+\sin\theta \cos\left(k\theta \right)\frac{\sin(\theta -\vartheta )}{\sin(k\theta )}\right)\\ \;=\frac{\sin\vartheta }{ak}. \end{array}$$Since $r_{+}\left(\theta \right)$ can be described as
$$r_{+}\left(\theta \right)=\frac{-b\left(\theta \right)+\sqrt{b{\left(\theta \right)}^{2}-4a\left(a-1\right)}}{2a}=-\frac{b(\theta )}{2a}\left(1+\sqrt{1-\frac{4a(a-1)}{b{\left(\theta \right)}^{2}}}\right),$$
we have
$$\begin{array}{ccc} \lim_{\theta \to \vartheta -0}r_{+}\left(\theta \right)\sin\left(\theta -\vartheta \right) & = & \lim_{\theta \to \vartheta -0}-\frac{b(\theta )\sin(\theta -\vartheta )}{2a}\left(1+\sqrt{1-\frac{4a(a-1)}{b{\left(\theta \right)}^{2}}}\right)\\ & = & -\frac{\sin\vartheta }{ak}. \end{array}$$Thus, we have
$$A=\left\{\begin{array}{cc} -\kappa \left(-\frac{\sin\theta_{0}}{a\kappa }+\sin\theta_{0}\right)=\left(\frac{1}{a}-\kappa \right)\sin\theta_{0} & (\mathrm{if}\;k=\kappa ),\\ -\kappa \left(-\frac{\sin\theta_{1}}{\kappa +1}+\sin\theta_{1}\right)=-\frac{\kappa^{2}}{\kappa +1}\sin\theta_{1} & (\mathrm{if}\;k=\kappa +1), \end{array}\right.$$
and, therefore, the proof is now complete. □

### 5.3. The Domain Ω for κ=∞

In this section, we deal with the case κ=∞. Since κθ(x,y) is regarded as *y* in this case, Equation (Equation 4) can be written as
(19)$$F\left(x,y\right)=x+\gamma x^{2}+\gamma y^{2}+y\cot y=0.$$

**Proposition** **3.**
*Assume that κ=∞ and let $x_{i}\left(y\right)$, i=1,2 be solutions of (19) with $x_{1}\left(y\right)\leq x_{2}\left(y\right)$ if they are real.*



(1)
*If γ=0, then one has $\Omega =\left\{z=x+yi;\;|y|<\pi \;\mathrm{and}\;x>-y\cot y\right\}$.*
(2)
*Suppose that $\gamma >\frac{1}{4}$. Then, there exists a unique function y=y(x) defined on R such that*

$$\Omega =\left\{z=x+yi\in C;\;x\in R\;\mathrm{and}\;|y|<y(x)\right\}.$$


*The function y(x) has a unique minimal point and tends to π as x→±∞.*
(3)
*If $0<\gamma \leq \frac{1}{4}$, then $x_{i}\left(y\right)$, i=1,2 are both real for any y∈(0,π), and one has*

$$\Omega =\left\{z=x+yi\in C;\;|y|<\pi \;\mathrm{and}\;x>x_{2}\left(y\right)\right\}.$$

(4)
*Suppose that γ<0. Then, there exists a unique $y_{0}\in \left(0,\pi \right)$ such that $x_{1}\left(y_{0}\right)=x_{2}\left(y_{0}\right)$ and if $y\leq y_{0}$ then $x_{i}\left(y\right)$, i=1,2 are real. Moreover, one has*

$$\Omega =\left\{z=x+yi\in C;\;|y|<y_{0}\;\mathrm{and}\;x_{1}\left(y\right)<x<x_{2}\left(y\right)\right\}.$$


*In particular, Ω is bounded.*



**Proof.** As with the proof of Proposition 2, we work with $\Omega^{+}=\Omega \cap C^{+}$. Let us assume y>0.(i) The case γ≤0 is dealt in the previous paper [1], and hence we omit it. Please note that The case γ=0 corresponds to the case of the classical Lambert function (cf. [19]).(ii) Assume that γ>0. Since there are no solutions of (Equation 19) on the line R+iπ and since Ω is symmetric with respect to the real axis, it is enough to consider *y* in the interval (0,π). The Equation (Equation 19) can be calculated as
(20)$$x^{2}+\frac{x}{\gamma }+y^{2}+\frac{y\cot y}{\gamma }=0\Leftrightarrow {\left(x+\frac{1}{2\gamma }\right)}^{2}=\frac{1}{4\gamma^{2}}-y^{2}-\frac{y\cot y}{\gamma }=:h\left(y\right).$$Let us set g(y):=cosy+2γysiny. Then, the function h(y) satisfies
$$h\left(y\right)=\frac{1-g{\left(y\right)}^{2}}{4\gamma^{2}\sin^{2}y},\;h\left(0\right)=\lim_{y\to 0}h\left(y\right)=\frac{1-4\gamma }{4\gamma^{2}}\;\mathrm{and}\;\lim_{y\to \pi -0}\left|h\left(y\right)\right|=+\infty .$$In order that Equation (Equation 20) has a real solution in *x* and *y*, the function h(y) needs to be non-negative, and it is equivalent to the condition that the absolute value of the function g(y) is less than or equal to 1. Therefore, we investigate the function g(y). At first, we observe that g(0)=1 and g(π)=−1. Its derivative is calculated as
$$\begin{array}{ccc} g^{\prime }\left(y\right) & = & -\sin y+2\gamma (\sin y+y\cos y)=-(1-2\gamma )\sin y+2\gamma y\cos y\\ & = & \left(2\gamma -1\right)\left(\frac{2\gamma }{2\gamma -1}y+\tan y\right)\cos y. \end{array}$$Set $c_{\gamma }=\frac{2\gamma }{2\gamma -1}$. Then, the signature of $g^{\prime }$ can be determined by the product of signatures of 2γ−1, cosy and $c_{\gamma }y+\tan y$.(ii-1) Assume that $\gamma >\frac{1}{4}$. If $\gamma >\frac{1}{2}$, then we have $c_{\gamma }>1$ and hence there exists a unique $y_{*}\in \left(\frac{\pi }{2},\pi \right)$ such that $c_{\gamma }y+\tan y=0$. If $\frac{1}{4}<\gamma <\frac{1}{2}$, we have $c_{\gamma }<-1$ so that there exists a unique $y_{*}\in \left(0,\frac{\pi }{2}\right)$ such that $c_{\gamma }y_{*}+\tan y_{*}=0$. Thus, for both cases, g(y) has a unique maximal point at $y=y_{*}$ with g(0)=1 and g(π)=−1. If $\gamma =\frac{1}{2}$, then we have $g^{\prime }\left(y\right)=y\cos y$, whence $g^{\prime }\left(y\right)>0$ for $y\in (0,\frac{\pi }{2})$ and $g^{\prime }\left(y\right)<0$ for $y\in (\frac{\pi }{2},\pi )$ so that in this case $y_{*}=\frac{\pi }{2}$ is the only solution of g(y)=0 for y∈(0,π).These observation shows that, if $\gamma >\frac{1}{4}$, then there exists one and only one $y_{0}\in \left(y_{*},\pi \right)$ such that $g(y_{0})=1$ and $g(y_{0}-\varepsilon )>1$ for $\varepsilon \in (0,y_{0}-y_{*})$. In this case, h(y) is non-negative in the interval $\left[y_{0},\pi \right)$, and $h(h_{0}-\varepsilon )<0$ for $\varepsilon \in (0,y_{0}-y_{*})$. Let $x_{i}\left(y\right)$, i=1,2 be the real solutions of Equation (Equation 20) with $x_{1}\left(y\right)\leq x_{2}\left(y\right)$. Then, since we have $x_{1}\left(y_{0}\right)=x_{2}\left(y_{0}\right)$, the curves $x_{i}\left(y\right)$, $y\in (y_{0},\pi )$ form a connected curve. Moreover, since the correspondence of $y\in (y_{0},\pi )$ to $x_{i}\left(y\right)$ is one-to-one and since $x_{1}\left(y\right)\to -\infty$, and $x_{2}\left(y\right)\to +\infty$ as y→π−0, the function y=y(x) connecting the two inverse functions of $x=x_{i}\left(y\right)$ is defined for any x∈R and its image is $\left[y_{0},\pi \right)$. Thus, we have
$$\Omega^{+}=\left\{z=x+yi\in C;\;x\in R\;\mathrm{and}\;0<y<y(x)\right\}.$$(ii-2) Assume that $0<\gamma \leq \frac{1}{4}$. Then, we have $-1\leq c_{\gamma }<1$ and hence there are no y∈(0,π) such that $c_{\gamma }y+\tan y=0$. Thus, we obtain $g^{\prime }\left(y\right)<0$ for any y∈(0,π) so that *g* is monotonic decreasing from g(0)=1 to g(π)=−1. This shows that h(y) is non-negative in the interval (0,π). Let $x_{i}\left(y\right)$, i=1,2 be the solutions of Equation (Equation 20) with $x_{1}\left(y\right)\leq x_{2}\left(y\right)$. Then, since $x_{1}\left(y\right)\to -\infty$ and $x_{2}\left(y\right)\to +\infty$ as y→π−0, we have by $x_{1}\left(0\right)<x_{2}\left(0\right)<0$
$$\Omega^{+}=\left\{z=x+yi\in C;\;0<y<\pi \;\mathrm{and}\;x>x_{2}\left(y\right)\right\},$$
which completes the proof. □

### 5.4. Bijectivity of $f_{\kappa ,\gamma }$

In this section, we investigate whether $f_{\kappa ,\gamma }$ maps $\Omega^{+}$ to $C^{+}$ bijectively, where $\Omega^{+}:=\Omega \cap C^{+}$. Then, since $f_{\kappa ,\gamma }\left(\bar{z}\right)=\bar{f_{\kappa ,\gamma }\left(z\right)}$, the bijectivity of $f_{\kappa ,\gamma }$ from Ω to $C\backslash \bar{S}$ is obtained at the same time.

The key tool is the argument principle (see Theorem 18, p.152 in [20], for example).

**Theorem** **4**
**(The argument principle).**
*If f(z) is meromorphic in a domain Ω with the zeros $a_{j}$ and the poles $b_{k}$, then*

$$\frac{1}{2\pi i}\int_{\gamma }\frac{f^{\prime }\left(z\right)}{f(z)}\;dz=\sum_{j}n\left(\gamma ,a_{j}\right)-\sum_{k}n\left(\gamma ,b_{k}\right)$$

*for every cycle γ which is homologous to zero in
Ω and does not pass through any of the zeros or poles. Here, n(γ,a) is the winding number of γ with respect to a.*


We also use the following elementary property of holomorphic functions.

**Lemma** **2.**
*Let f(z)=u(x,y)+iv(x,y) be a holomorphic function. The implicit function v(x,y)=0 has an intersection point at z=x+yi only if $f^{\prime }\left(z\right)=0$.*


#### 5.4.1. The Case κ≥1 and γ<0

This case is made in the previous paper [1], and thus we omit it.

#### 5.4.2. The Case κ≥1 and γ≥0

In this case, Propositions 1–3 tell us that Ω is unbounded. We first consider S defined in (Equation 6). Please note that the case (κ,γ)=(1,1) is the trivial case $f_{\kappa ,\gamma }\left(z\right)=z$.

**Lemma** **3.**
*Let $\alpha_{i}$, i=1,2 be the solutions of (9).*



(1)
*Assume that κ>1 or κ=∞ together with D(0)≥0. Then, $\alpha_{i}$ are both real and $S=(-\infty ,f_{\kappa ,\gamma }\left(\alpha_{2}\right))$ with $f_{\kappa ,\gamma }\left(\alpha_{2}\right)<0$. Please note that γ=0 is included in this case.*
(2)
*Assume that κ=1 and 0<γ<1. Then, $\alpha_{i}$ are both real, and one has $S=(f_{\kappa ,\gamma }\left(\alpha_{1}\right),f_{\kappa ,\gamma }\left(\alpha_{2}\right))$ with $f_{\kappa ,\gamma }\left(\alpha_{2}\right)<0$. Please note that 0<γ<1 is equivalent to D(0)>0.*
(3)
*Assume that κ≥1 or κ=∞ together with D(0)<0. Then, $\alpha_{i}$ are both non-real, and one has S=∅. On the other hand, one has $f_{\kappa ,\gamma }\left(\partial \Omega \cap C^{+}\right)=\left(-\infty ,0\right)$.*



**Proof.** The proof is elementary, so we only deal with the latter assertion in (3). It is easily verified that $|f_{\kappa ,\gamma }\left(z\right)|\to +\infty$ as |z|→+∞ by κ>1. Propositions 1–3 show that if a point $z=x+yi\in C^{+}$ is on the curve $\partial \Omega \cap C^{+}$, then we have F(x,y)=0 and
(21)$$f_{\kappa ,\gamma }\left(z\right)=\frac{{\left|1+\frac{z}{\kappa }\right|}^{\kappa }}{{\left|1+\gamma z\right|}^{2}}\cdot \frac{-y}{\sin(\kappa \theta (x,y))}<0.$$
This means that if $z\in C^{+}$ goes to *∞* along the path $\partial \Omega \cap C^{+}$, then $f_{\kappa ,\gamma }$ must tend to −∞. On the other hand, taking a limit y→+0 along $\partial \Omega \cap C^{+}$ is equivalent to z→−κ in this case and hence $f_{\kappa ,\gamma }\left(z\right)\to 0$. Since $f_{\kappa ,\gamma }$ is continuous, the assertion is now proved. □

For L>0, let $\Gamma_{L}$ be the circle $-\kappa +Le^{i\theta }$ of origin z=−κ with radius *L*. We distinguish cases according to this lemma.

*Case (1).* In this case, Ω is given in Propositions 2 or 3. Let us take a path C=C(t), t∈(0,1) in such a way that by starting from z=∞, it goes to $z=\alpha_{2}$ along the curve $r_{+}$ describing $\partial \Omega \cap C^{+}$ in the upper half plane, and then goes to z=∞ along the real axis. Here, we can assume that $C^{\prime }\left(t\right)\neq 0$ whenever $C\left(t\right)\neq \alpha_{i}$, i=1,2 because $f_{\kappa ,\gamma }^{\prime }\left(C\left(t\right)\right)\neq 0$ otherwise. We take an arc-length parameter *t* so that $C^{\prime }\left(t\right)$ represents the direction of the tangent line at x+yi=C(t). Lemma 2 tells us that $g\left(t\right):=f_{\kappa ,\gamma }\left(C(t)\right)$ is a monotonic increasing function on (0,1) such that g(t)→−∞ as t→0 and g(t)→+∞ as t→1.

We shall show that for any $w_{0}\in C^{+}$ there exists one and only one $z_{0}\in C^{+}$ such that $f_{\kappa ,\gamma }\left(z_{0}\right)=w_{0}$. Recall that Imfκ,γ′(z)>0 for any $z\in C^{+}$. Let us take an R>0 such that $|w_{0}|<R$. For an L>0, let $z_{i}$, i=1,2 be two distinct intersection points of *C* and $\Gamma_{L}$. Please note that we can take $z_{1}=-\kappa +L\in R$ and $z_{2}\in C^{+}$. Let $\tilde{C}$ be a closed path obtained from *C* by connecting $z_{1}$ and $z_{2}$ via the arc $A_{L}$ of $\Gamma_{L}$ included in $C^{+}$.

Let ${\tilde{\Omega }}^{+}$ be the inside set of $\tilde{C}$. Since $f_{\kappa ,\gamma }$ is non-singular on the arc $A_{L}$, the curve $f_{\kappa ,\gamma }\left(A_{L}\right)$ does not have a singular point so that it is homotopic to a semicircle in $C^{+}$. In particular, we can take an *L* such that its radius is larger than *R* so that the inside set $f_{\kappa ,\gamma }\left({\tilde{\Omega }}^{+}\right)$ of the curve $f_{\kappa ,\gamma }\left(\tilde{C}\right)$ is a bounded domain including $w_{0}\in C^{+}$. Since the winding number of the path $f_{\kappa ,\gamma }\left(\tilde{C}\right)$ about $w=w_{0}$ is exactly one, we see that
$$\frac{1}{2\pi i}\int_{\tilde{C}}\frac{f_{\kappa ,\gamma }^{\prime }\left(z\right)}{f_{\kappa ,\gamma }\left(z\right)-w_{0}}dz=\frac{1}{2\pi i}\int_{f_{\kappa ,\gamma }\left(\tilde{C}\right)}\frac{dw}{w-w_{0}}=1.$$Since $f_{\kappa ,\gamma }$ does not have a pole on ${\tilde{\Omega }}^{+}$ by definition, the function $f_{\kappa ,\gamma }\left(z\right)-w_{0}$ has only one zero point, say $z_{0}$, by the argument principle in Theorem 4. Then, we obtain $f_{\kappa ,\gamma }\left(z_{0}\right)=w_{0}$, and such $z_{0}$ is unique. Since we can take $w_{0}\in C^{+}$ arbitrary, we conclude that the map $f_{\kappa ,\gamma }$ is a bijection from $\Omega^{+}$ to the upper half plane $C^{+}$.

*Case (2).* In this case, Ω is given in Proposition 1 (2). As in the case (1), it is enough to show that an approximating domain ${\tilde{\Omega }}^{+}$ of $\Omega^{+}=\Omega \cap C^{+}$ satisfies that the image of its boundary curve $\tilde{C}=\partial {\tilde{\Omega }}^{+}$ under $f_{\kappa ,\gamma }$ has a winding number about $w=w_{0}$ being exactly one for any $w_{0}\in C^{+}$.

By the proof of Proposition 1, the boundary of $\Omega^{+}$ can be described as a path C(t), t∈(0,1) defined in a such a way that by staring z=∞, it goes to $z=\alpha_{1}$ along the real axis, and then it goes to $z=\alpha_{2}$ along the curve $\partial \Omega \cap C^{+}$, which is a semicircle of origin $z=-\frac{1}{\gamma }$ with radius $\sqrt{1-\gamma }/\gamma$, and then it goes to z=∞ along the real axis. Notice that $f_{\kappa ,\gamma }^{\prime }\left(\tilde{C}\left(t\right)\right)\neq 0$ for any t∈(0,1) except for $t_{i}$ such that $C\left(t_{i}\right)=\alpha_{i}$, i=1,2. In particular, Lemma 2 tells us that $f_{\kappa ,\gamma }\left(C\left(t\right)\right)$ is monotonic increasing in the interval $\left(t_{1},t_{2}\right)$, and hence the function $f_{\kappa ,\gamma }\left(C\left(t\right)\right)$ maps (0,1) to R bijectively. Therefore, we can construct an approximating domain ${\tilde{\Omega }}^{+}$ from C(t) and $\Gamma_{L}$ similar to case (1), such that the image of its boundary $\tilde{C}$ under $f_{\kappa ,\gamma }$ has a winding number being exactly one, and hence we have shown the bijectivity of $f_{\kappa ,\gamma }$ on $\Omega^{+}$ to $C^{+}$.

*Case (3).* In this case, we shall show that the approximating domain ${\tilde{\Omega }}^{+}$ of $\Omega^{+}=\Omega \cap C^{+}$ satisfies that the image of its boundary curve $\tilde{C}=\partial {\tilde{\Omega }}^{+}$ under $f_{\kappa ,\gamma }$ has a winding number about $w=w_{0}$ being exactly two for any $w_{0}\in C^{+}$, and hence the function $f_{\kappa ,\gamma }$ maps $\Omega^{+}$ to $C^{+}$ two-to-one.

(i) Let us assume that κ≥1. The domain Ω is given in Proposition 2 (5-b). Let C(t), t∈(0,1) be a path defined in such a way that by starting from z=∞, it goes to z=−κ along the curve $\partial \Omega \cap C^{+}$, and then goes to z=+∞ along the real axis. Let $t_{0}\in \left(0,1\right)$ such that $C(t_{0})=-\kappa$. Then, Lemma 3 (3) tells us that $f_{\kappa ,\gamma }\left(C\left(t\right)\right)$ maps $\left(0,t_{0}\right)$ to (−∞,0) bijectively, and hence we see that it maps (0,1) to R in two-to-one correspondence. Let ${\tilde{\Omega }}^{+}=\Omega^{+}\cap \Gamma_{L}$ and $\tilde{C}$ its boundary. the image of its boundary curve $\tilde{C}=\partial {\tilde{\Omega }}^{+}$ under $f_{\kappa ,\gamma }$ has a winding number about $w=w_{0}$ being exactly two for any $w_{0}\in C^{+}$, and hence the function $f_{\kappa ,\gamma }$ maps $\Omega^{+}$ to $C^{+}$ two-to-one.

(ii) Assume that κ=∞ together with D(0)<0, i.e., $\gamma >\frac{1}{4}$. Then, Ω is described in Proposition 3 (2). Take a large R>0 and a small ε>0 such that $\varepsilon <|w_{0}|<R$. Then, we approximate $\Omega^{+}$ by the curve $\tilde{C}\left(t\right)$, t∈[0,1] defined in such a way that for L>0 and δ>0, by starting from $\tilde{C}\left(0\right)=-L$, it goes to z=L along the real axis avoiding $z=-\frac{1}{\gamma }$ along a semicircle of radius δ>0 in $C^{+}$, and next to z=L+y(L)i along the line parallel to the imaginary axis, and next to z=−L+y(−L)i along $\partial \Omega \cap C^{+}$ and finally goes back to $\tilde{C}\left(1\right)=-L$ along the line parallel to the imaginary axis. Then, since $|f_{\infty ,\gamma }\left(z\right)|=\left|\frac{z}{1+\gamma z}\right|e^{x}=\frac{e^{x}}{|\gamma +\frac{1}{z}|}$, we can choose L>0 such that
$$\left\{\begin{array}{c} |f_{\infty ,\gamma }\left(-L+yi\right)|<\varepsilon \;\left(y\in \left(0,y\left(-L\right)\right)\right),\\ |f_{\infty ,\gamma }\left(L+yi\right)|>R\;\left(y\in \left(0,y\left(L\right)\right)\right), \end{array}\right.$$
and since |1+γz|=γδ and $ze^{z}$ are bounded around $z=-\frac{1}{\gamma }$, we can choose δ>0 such that $\left|f_{\infty ,\gamma }\left(-\frac{1}{\gamma }+\delta e^{i\theta }\right)\right|>R$. Since $f_{\infty ,\gamma }$ is non-singular on the lines ±L+yi, y∈(0,y(±L)), the curve $f_{\infty ,\gamma }\left(C^{\prime }\left(t\right)\right)$, t∈[0,1] contains the domain obtained from a semicircle of radius *R* removing a semicircle of radius ε. In particular, it contains $w_{0}$. Since we can show that the winding number of the path $f_{\infty ,\gamma }\left(\tilde{C}\right)$ with respect to $w=w_{0}$ is exactly two, we conclude that the function $f_{\kappa ,\gamma }$ maps $\Omega^{+}$ to $C^{+}$ in a two-to-one way by a similar argument as in (i) above.

#### 5.4.3. The Case 0<κ<1 and γ≤0

Although this case is out of the result stated in the previous paper [1], the same proof given there (see the supplementary material of [1]) is valid, and therefore we omit it.

#### 5.4.4. The Case 0<κ<1 and γ>0

In this case, Proposition 2 tells us that if 0<a=κγ<1 then $\Omega \cap R=(\alpha_{2},+\infty )$, and if a>1 then $\Omega \cap R=\left\{x\in R;\;x>-\kappa \;\mathrm{and}\;x\neq -\frac{1}{\gamma }\right\}$. If $f_{\kappa ,\gamma }$ maps $\Omega^{+}$ to $C^{+}$ bijectively, then it needs map $\partial \Omega^{+}$ to R. However, if x∈R satisfies $x<\min(\alpha_{1},-\kappa )$, then $\exp_{\kappa }\left(z\right)$ tends to ${\left|1+\frac{x}{\kappa }\right|}^{\kappa }\;e^{i\kappa \pi }$ as z→x via the arc $re^{i\theta }=1+\frac{z}{\kappa }$. Since 0<κ<1, then we see that $\lim_{z\to x}f_{\kappa ,\gamma }\left(z\right)$ is not real and therefore $f_{\kappa ,\gamma }$ cannot map $\Omega^{+}$ to $C^{+}$ bijectively.

#### 5.4.5. The Case of κ<0

We shall complete the proof of Theorem 3 by proving it for the case κ<0. To do so, let us recall the homographic (linear fractional) action of SL(2,R) on C: $g\cdot z:=\frac{az+b}{cz+d}$ for $g=\left(\begin{array}{cc} a & b\\ c & d \end{array}\right)\in SL\left(2,R\right)$ and $z\in C^{+}$. For each g∈SL(2,R), the corresponding homographic action map $C^{+}$ to $C^{+}$ bijectively. Let $\kappa =-\kappa^{\prime }$ with positive $\kappa^{\prime }>0$. Consider the transformation
$$1+\frac{z^{\prime }}{\kappa^{\prime }}={\left(1+\frac{z}{\kappa }\right)}^{-1}.$$Notice that this transformation can be written as a homographic action as $z^{\prime }=\left(\begin{array}{cc} 1 & 0\\ 1/\kappa & 1 \end{array}\right)\cdot z$, and hence it maps $C^{+}$ to $C^{+}$ bijectively. Then, since
$$\frac{z}{1+\gamma z}=\frac{z^{\prime }}{1+\left(\gamma +1/\kappa^{\prime }\right)z^{\prime }}\;\mathrm{and}\;{\left(1+\frac{z}{\kappa }\right)}^{\kappa }={\left({\left(1+\frac{z}{\kappa }\right)}^{-1}\right)}^{-\kappa }={\left(1+\frac{z^{\prime }}{\kappa^{\prime }}\right)}^{\kappa^{\prime }}$$(recall that we are taking the main branch so that $\log z=-\log(z^{-1})$), we obtain
$$f_{\gamma ,\kappa }\left(z\right)=\frac{z}{1+\gamma z}{\left(1+\frac{z}{\kappa }\right)}^{\kappa }=\frac{z^{\prime }}{1+\left(\gamma +1/\kappa^{\prime }\right)z^{\prime }}{\left(1+\frac{z^{\prime }}{\kappa^{\prime }}\right)}^{\kappa^{\prime }}=f_{\gamma +1/\kappa^{\prime },\;\kappa^{\prime }}\left(z^{\prime }\right).$$Set $\gamma^{\prime }=\gamma +1/\kappa^{\prime }$. Since homographic actions map $C^{+}$ to $C^{+}$ bijectively, there exists a domain Ω such that $f_{\kappa ,\gamma }$ maps $\Omega^{+}=\Omega \cap C^{+}$ to $C^{+}$ bijectively if and only if it holds for $f_{\gamma^{\prime },\kappa^{\prime }}$. Thus, the condition $\gamma^{\prime }\leq 0$ is equivalent to $\gamma \leq \frac{1}{\kappa }$, and the condition $\kappa^{\prime }>1$ and $\gamma^{\prime }>0$ with $\gamma^{\prime }\leq \frac{1}{4}{\left(1+\frac{1}{\kappa^{\prime }}\right)}^{2}$ is equivalent to
$$\gamma >\frac{1}{\kappa }\;\mathrm{and}\;\kappa <-1\;\mathrm{with}\;\gamma -\frac{1}{\kappa }\leq \frac{1}{4}{\left(1-\frac{1}{\kappa }\right)}^{2}\Leftrightarrow \gamma \leq \frac{1}{4}{\left(1+\frac{1}{\kappa }\right)}^{2}.$$This shows the case κ<0, and, therefore, we have completed the proof of Theorem 3.

## Figures and Tables

**Figure 1 entropy-25-00858-f001:**
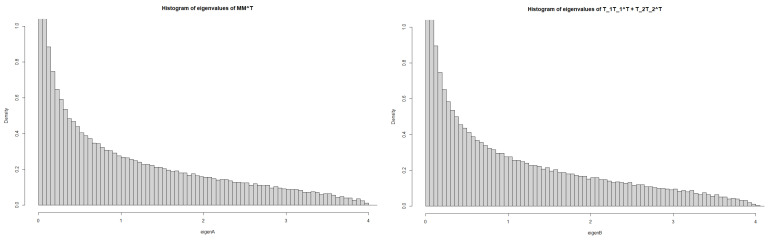
The left one indicates the histogram of eigenvalues of $MM^{⊤}$, whereas the right that of $T_{1}T_{1}^{⊤}+T_{2}T_{2}^{⊤}$.

**Figure 2 entropy-25-00858-f002:**
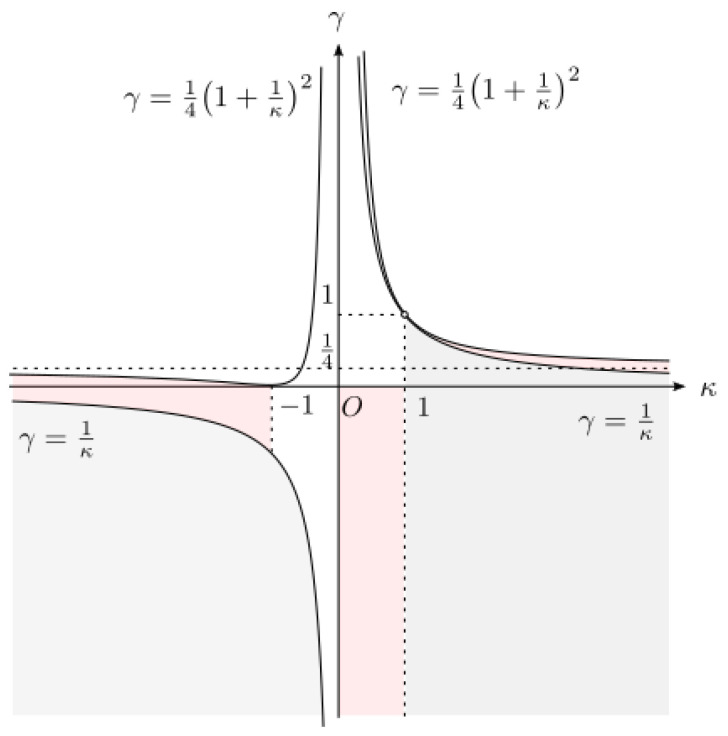
The parameter domain on κ and γ.

**Table 1 entropy-25-00858-t001:** Monotonicity table of $F_{\kappa }\left(\theta \right)$.

Range of κ	Increasing/Decreasing Table	Note
0<κ≤12	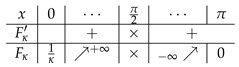	
12<κ<1	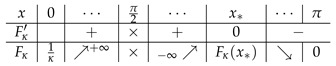	Fκ(x*)<1
1<κ<2	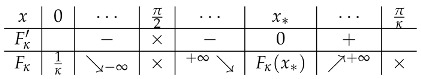	Fκ(x*)>1
κ≥2	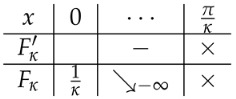	

In the table, the symbol × means that the functions are not defined at that point. The symbol $x_{*}$ denotes a maximal/minimal point in the interval $I_{0}$, if it exists.

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
