# Peer review of "Stieltjes Transforms and R-Transforms Associated with Two-Parameter Lambert–Tsallis Functions"

_entropy, 2023, doi:10.3390/e25060858_

Round 1

Reviewer 1 Report

The paper is devoted to the study of the behavior of the sample covariance matrix spectrum in the case when the sample size is equal to the number of observed parameters, i.e. the matrices of observations is square, in other words, to the study of the quarter circular law in the local regime. The paper "Stiltjes transforms and $R$-transforms related to two-parameter Lambert-Tsallis functions" is for a wide range of readers. The paper deals with Winberg matrices, more precisely Winberg cone - this class includes well-known matrix ensembles - Wigner matrices and sampling matrices. We investigate representations for Stiltjes transformations and $R$-transformations of the spectral measure of such matrices. The paper is a continuation of the paper "Wigner and Wishart ensembles for sparse Vinberg models" by the same authors. The authors use all notations of the last paper. For example, formula (1.2) uses the notation $Mat(n,N;\mathbb R)$, which is not described. Sometimes the notation $Mat(a_n\times b_n; \mathbb R)$ described only in the article "Wigner and Wishart ensembles for sparse Vinberg models" is used. I think it is better to describe all models and notations before their first occurrence for the reader's convenience. At the very least, provide an electronic link to the article "Wigner ..." so that the reader can easily find the descriptions he needs.   I strongly recommend the article by Nakashima and Graczuk to be published in the special issue "Random matrices: Theory and Applications" .

Round 2

Reviewer 2 Report

The authors corrected a number of issues I raised in my initial report. I am happy with the final result and do not objet to the pubilcation of this preprint in Entropy